# Process Parameter Investigation and Torsional Strength Analysis of the Additively Manufactured 3D Structures Made of 20MnCr5 Steel

**DOI:** 10.3390/ma16051877

**Published:** 2023-02-24

**Authors:** Bartłomiej Sarzyński, Janusz Kluczyński, Jakub Łuszczek, Krzysztof Grzelak, Ireneusz Szachogłuchowicz, Janusz Torzewski, Lucjan Śnieżek

**Affiliations:** Institute of Robots & Machine Design, Faculty of Mechanical Engineering, Military University of Technology, Gen. S. Kaliskiego St., 00-908 Warsaw, Poland

**Keywords:** additive manufacturing, torsional strength, cellular structures, 20MnCr5 Steel

## Abstract

An ongoing growth of the available materials dedicated to additive manufacturing (AM) significantly extends the possibilities of their usage in many applications. A very good example is 20MnCr5 steel which is very popular in conventional manufacturing technologies and shows good processability in AM processes. This research takes into account the process parameter selection and torsional strength analysis of AM cellular structures. The conducted research revealed a significant tendency for between-layer cracking which is strictly dependent on the layered structure of the material. Additionally, the highest torsional strength was registered for specimens with a honeycomb structure. To determine the best-obtained properties, in the case of the samples with cellular structures, a torque-to-mass coefficient was introduced. It indicated the best properties of honeycomb structures, which have about 10% smaller torque-to-mass coefficient values than monolithic structures (PM samples).

## 1. Introduction

Additive manufacturing (AM) technologies allow the production of cellular structures, which cannot be made by other manufacturing technologies. They make it possible to achieve advantageous performance characteristics for the components and machine elements in which they are used. The main benefit of their use is their low weight, positively determining the performance generated by the device. Another important factor in favor of their use is the reduction in the amount of material used, which leads to a reduction in the cost of manufacturing the component. The ongoing growth of AM technologies requires providing significant knowledge of the behavior of the material during various types of loading and the production of cellular structures. Additionally, one of the most important advantages, based on an in situ performance property modification [1,2,3], is that it allows for the production of final-use parts with enhanced parameters, which are not possible to obtain with the use of conventional manufacturing technologies, i.e., molds, or drills with conformal cooling [4,5]. Furthermore, additive technology has recently shown promise for the production of components with alloys that cannot be processed with traditional processes, for example in the case of magnetic FeSi steels with a high Si content [6,7,8]. Many AM parts are subjected to torsional loads [9,10,11,12], which is not a very popular path in the available literature, in comparison to tensile or bending strength analysis. Such analysis is very important, especially in the case of steel powders dedicated to manufacturing high-loaded parts subjected to significant torque values. To this group of parts belong i.a. gears and shafts. Researchers have tried to implement a lattice structure to the body of these parts [13,14]. However, to generate an appropriate structure, it is important to know how the cellular structures carry the load in relation to the solid material. In the present state of the art, some works are related to the torsional strength analysis of AM parts. Halama [15] et al. showed a negative effect of additional postprocessing (machining) during torsional tests on specimens produced with the use of laser-based powder bed fusion of metal (PBF-LB/M) technology. In their research, “as built” specimens showed approximately 2/3 higher ductility than machined counterparts. The authors claimed that “as built” parts were characterized by ductile fracture when at the same time the machined specimen only had the local occurrence of ductile fractures and the prevailing features of brittle fractures. Macek et al. [16], during their analysis related to fatigue torsional tests of 18Ni300 steel, revealed a significant influence of an interlayer defect, near the surface, from which the crack nucleated into the volume of the test part. In the case of using high-pressure torsion, Han et al. [17] registered the excess number of dislocations during the nanostructuring of AM 316L stainless steel (increase by 0.27–0.32%). In another study related to torsional fatigue tests of Ti6Al4V, the authors [18] showed the shorter torsional fatigue lives of Ti6Al4V compared to the wrought material, when comparing shear strain amplitude. At the same time, AM parts were characterized by decreased cyclical softening during fatigue testing in comparison to wrought counterparts. What is more, it was found that additional annealing improves the torsional fatigue life of AM parts by more than an order of magnitude. Such a phenomenon was described by Fatemi et al. [18] and was related to the detrimental tensile residual stress relaxation generated during the AM manufacturing process, and perhaps due to increased local ductility at pores. In different research related to torsional strength analysis, Mirone et al. [19] highlighted a significant influence of surface roughness on Ti6Al4V specimens subject to torsional strength tests and registered a non-homogeneous ductile matrix mixed with brittle textures in the fracture analysis; this is a condition which reveals substantial embrittlement in fracture behavior, if compared to the static tensile loading.

In the present state of the art, there is a small amount of research related to the torsional strength analysis of AM parts obtained with the use of PBF-LB/M technology. Additionally, the availability of new steels dedicated to some exact applications (i.e., 20MnCr5 steel dedicated to carbonizing heat treatment) encourages the research of such alloys. In this research, AM lattice structures made of 20MnCr5 steel were taken into account and subjected to torsional strength tests. Such an analysis was conducted to determine material behavior during the torsional loading of thin-walled specimens.

## 2. Materials and Methods

Using SolidWorks 2021 software (Dassault Systems; Waltham, QC, Canada), CAD parts of test samples, including monolithic material (Figure 1), and two types of cell structures (Figure 2 and Figure 3) were made. All specimens’ geometries were based on the ISO18338:2015 standard. As a reference, a monolithic specimen was compared with all the obtained test results, the torsional strength of the material was calculated, and the most advantageous solution in terms of the strength-to-weight ratio of each component was identified.

The second geometry was based on a “Kagome” structure [20] with cell diameters equal to 0.6 mm, 0.8 mm, and 1 mm. The shape of a single cell and the whole specimen is shown in Figure 2. The part was designed with a fully open structure, allowing the unmelted material to be easily removed.

A 20MnCr5 steel powder was used to make the test specimens. It is a gas-atomized, low-alloy structural steel suitable for surface hardening, which gives it a high hardness of the surface layer while allowing the core of the component to remain plastic. Table 1 shows the powder’s particle diameter and the chemical composition of the steel.

Based on our own previous research [21] on steel from the same group of alloys (21NiCrMo2), it was possible to design a parameter development matrix for 20MnCr5. Both materials are case-hardened steels with comparable chemical compositions. Therefore, it was decided to try to use similar process parameters. Moreover, low-alloy steels used in additive manufacturing are characterized by similar ranges of process parameters, regardless of the steel used [22,23,24]. The crucial process parameter values were the exposure velocity, laser power, and the hatching distance (clearance between exposure paths). The thickness of the powder layer was fixed with a constant value. With the component data, it was possible to calculate the energy per unit volume. For this purpose Equation (1) was used. Samples manufactured for the selection of printing parameters are shown in Figure 4. Table 2 shows the 28 groups of printing parameter values assigned to each specimen.
(1)$$E_{V}=\frac{P_{L}}{h*v_{s}*H}$$

$E_{V}$—energy per unit volume (J/mm^3^)

$P_{L}$—laser power (W)

h—hatching distance (mm)

$v_{s}$—exposure velocity (mm/s)

H—thickness of powder layer (mm)

The Keyence (Osaka, Japan) VHX-7000 digital microscope (Figure 5) was used to analyze the quality of the sample structures that were produced using different printing parameters. Twenty-eight samples in two planes (parallel and perpendicular to layers deposition direction) were analyzed with this method. Such an approach allowed us to show the structure of the material in the melt plane (XY plane) and through the successive layers (YZ plane). The microscopic analyses for each sample were divided into two stages. In the first step, stitching of sequentially exposed areas was performed, and then the ratio of the porous area detected through the optical device to the total area of the sample was determined. The measurements for all specimens were performed using the same settings related to the detection of defects. The porosity value presented in this manuscript is based on a single measurement for each plane. The selection of the appropriate group of production parameters was made on the basis of the analysis of the results of porosity measurements. The group of parameters that allowed the production of the model element with the smallest number of pores was selected as the group used in the production of torsion test specimens.

Based on the analysis of the samples’ structure produced with different parameters, a group of samples with the lowest porosity was selected. During the AM process, it was planned to produce three specimens of each type:monolithic sample—MS,sample containing honeycomb structure—H,sample containing a bar structure with a bar diameter of 0.6 mm—S06,sample containing a bar structure with a bar diameter of 0.8 mm—S08,sample containing a bar structure with a bar diameter of 1 mm—S1.

All samples (shown in Figure 6) were produced in the vertical direction.

The final stage of research was to carry out torsion tests on the manufactured specimens. For this purpose, we used the Instron 8802 testing machine presented in Figure 7 was used. The testing machine was equipped with an additional torsion-testing module designed to perform static and fatigue tests.

By the means of the software integrated with the testing machine, it was possible to determine the precise character of the torsion process. The specimens were loaded at a rotational speed equal to 1°/s, (0.017 rad/s). Measuring sensors were placed in the rotating parts to register the load values of the specimen with an accuracy of 0.001 Nm. The data sampling frequency was 100 Hz.

## 3. Results and Discussion

### 3.1. AM Process Parameters Selection

The selection of the appropriate group of parameters for the production process was based on the analysis of the porosity of the samples produced with their use. Figure 8 shows an example of porosity analysis with the automatic detection of voids. This allowed the system to calculate the area identified as porous to the total measured zone. Some black dots/areas with a shade on the photo were recognized as sources of oxidation and pollution. This steel in the state as built is very susceptible to corrosion. These areas are not included in the porosity measurements.

Depending on the parameters used in the manufacturing process, the appearance of the specimens was significantly different. Figure 9 shows a specimen for which the twenty-seventh parameter group was used during the AM process. The laser power was 255 W, the exposure speed was 800 mm/s, and the hatching distance spacing was 0.12 mm. Based on Equation (1), the energy per unit volume was calculated and its value was 88.5 J/mm^3^.

Analyzing the images of the sample, a significant surface area of the porous part becomes visible immediately. The black, irregular spots indicate voids in the surface of the sample. Such defects have a very negative effect on the mechanical properties of components manufactured using a given series of parameters. During the transfer of loads through the components, stresses may be promoted in the areas concerned or notch effects may occur, leading to the formation of cracks and further material cracking. These observations were confirmed by carrying out a computer evaluation of the porous surface, which is visible in Figure 10. The surface area of defects in the sample was 3.85% in the XY plane and 4.48% in the YZ plane, giving an average of 4.17%.

The minimum porous surface area of both analyzed cross-sections was registered for the 2nd parameter’s group (shown in Figure 11). Process parameters in this group were as follows: a laser power equal to 225 W, an exposure velocity of 600 mm/s, and a hatching distance of 0.10 mm. The energy per unit volume was 125 J/mm^3^. There are significant differences between this measurement series and the previous ones. Based on the Keyence VHX-7000 microscope software’s indication of the pore areas, the number of pores was small enough to accept it (below 0.5%). This resulted in a porosity of only 0.06% in the XY plane and 0.26% in the YZ plane, giving an overall average of 0.16%. It can therefore be concluded that these parameters are appropriate for the material in question and their use will allow components with a very good quality structure to be made.

In order to show more clearly which samples had the highest and lowest porosity values, a bar graph is shown in Figure 12. It consists of two data series. The blue colors show the porosity values of the samples in the XY plane, while the orange colors show the porosity in the YZ plane. It is possible to read the highest porosity for samples: 7, 8, 22, and 27. The smallest porosity was registered for samples: 1, 2, 14, and 15.

Based on an analysis of the specimen structure manufactured with different parameters, a group of those that provided the lowest porosity was selected. Their values are shown in Table 3.

### 3.2. Torsion Tests

The torsion tests were started for monolithic specimens. Differences between the torsion angle of PM#1 and the other two cases were caused by fractured specimens PM#2 and PM#3A. The sample marked by PM#1 did not fracture during the test; the machine reached the maximum level of the torsion angle. The total rotation made by the moving part of the testing machine was 61°. At a given point, the loading torque on the specimen was 172 Nm. The stresses for all tested samples were calculated and shown in Figure 13. The similar behavior of all samples can be seen. Each line plotted through the numerical values recorded by the measuring devices is very similar. In the range of torsion angle 0–15°, the specimens deform in the elastic range, exceeding 15°, and they reach a plastic state (characteristic “flattening” of the data line). Based on the obtained results, it can be concluded that the material has a fairly low torsional strength limit.

The torsion test of the S06 specimen series indicated a similar behavior of the test samples in all three cases. Due to their very thick structure, the specimens broke at quite small angles. Specimen S06#1 failed at a torque of 9.56 Nm and a torsion angle of 4.37°. Sample S06#2 achieved the most favorable result of 9.79 Nm at an angle of 6.97°. The third specimen fractured under a torsional moment of 9.86 Nm, twisting by an angle of 6.17°. All changes in the material occurred in terms of elastic deformation. The results are shown in Figure 14. The difference between the smallest and largest value of a given parameter is only 0.3 Nm. This demonstrates the consistent failure mechanism of the specimens manufactured.

Sample S08#1 reached an angle of 7.3° at a torsional torque of 21 Nm. Specimen S08#2 failed at the highest value of the torsional moment. In this case, its value was 21.6 Nm, which corresponds to an angle of 7.8°. The last case retained the lowest torsion strength. Specimen S08#3 broke at an angle of 7.5° with a torsional moment of 20.3 Nm. All curves are similar to each other which indicates repeatability of all parts’ properties. The angle of twist was between 7.3° and 7.8° with a torsional torque of 21 Nm and 21.6 Nm. The results are shown in Figure 15.

The last of the bar structures tested was sample S1. The test results were included in the torque–torsion angle relationship shown in Figure 16. In this case, specimen S1#1 reached a torsion angle of 7.82°, breaking under a torsional torque of 38 Nm. This case achieved the highest values of the parameters described. Specimen S1#2 broke, reaching an angle of 7.1° under a torsional torque of 37 Nm. Specimen S1#3 had the lowest strength due to the torsional moment. The specimen broke, reaching 36 Nm at a torsion angle of 7.6°.

As a final case of cellular structure, a structure with a geometry of contacting hexagons, i.e., a “honeycomb” resemblance was tested. The test results are shown in Figure 17. In the given test series, specimen HS#1 achieved a torsion angle of 20.3° at a torsional torque of 63.8 Nm. Specimen HS#2 broke at the same torque and an angle of 18.7°. The last specimen resisted the highest load of 64.5 Nm and at an angle of 20.8°. Analyzing the graph, it is apparent that the individual data series are very similar. In the 0–8° range, all of the specimens deform elastically. When exceeding an angle value equal to 8°, the specimens indicated plastic behavior. The maximum value of the torsional moment in all three cases is strongly similar to each other.

To achieve the main aim of this research project, the determination of the most favorable cell structure in terms of low mass and high strength was carried out. To this aim, a special parameter was determined using Formula (2) with the assumed torque-to-mass coefficient “*W_M_*” being determined.
(2)$$W_{M}=\frac{M_{S}}{m_{s}}$$


$W_{M}$—torque-to-mass coefficient,
$M_{S}$—torque at which the specimen was broken,
$m_{s}$—a mass of specimen structure.


The results of the calculations are summarized in Table 4 and the chart is shown in Figure 18.

Table 4 and the columnar diagram shown in Figure 18 allow us to make several observations. Of all the samples tested, the monolithic sample has the highest W_M_ value. The cellular structure with the highest W_M_ coefficient is the honeycomb structure. The highest value of the given parameter is found in sample HS#3 and is 4828 Nm/kg. Furthermore, these samples have the second smallest mass of all structure cases. Considering the samples with a “Kagome” structure, a certain relationship is apparent. As the mass of the sample increases, the W_M_ factor also increases. Sample S06#1 has the lowest W_M_ coefficient of all cases—its value is equal to 904 Nm/kg. The highest value of the described parameter among the specimens with a “Kagome” structure is found in the case of S1#1 and in the case of S1#1 is 1435 Nm/kg. Furthermore, the unit cell of the honeycomb structure has a privileged orientation related to torsional load, in contrast to the “Kagome” structure. Additionally, the area of a cross-section of the honeycomb structure is uniform throughout the whole heigh samples. The Kagome sample area of the cross-section is changeable. The PS samples always crack in the area of the cross-section that is the smallest. Figure 19 shows a chart with the dependence of torsional moment on the torsion angle for the specimens that had the highest strength in their groups. A large difference can be seen between the torsional moment values of the most favorable cellular specimens and the monolithic specimens. In this case, the cellular structures are no match for the monolithic sample. Nevertheless, the other advantages of the structures must be taken into account. When comparing the mass and the amount of material used in the manufacture, the obtained results are more favorable for samples containing cellular structures.

Figure 20 presents the fractures of the tested samples. In Figure 20a, red dashed lines highlight the areas of plastic deformation of the material. The yellow color highlights the area of brittle fracture of the sample, part of which can be seen in the magnification on the right. In the case of the honeycomb structure (Figure 20b), the fracture is brittle in nature, and the specimen geometry itself has not been deformed significantly. The fractures of the specimens containing the Kagome-type structure show the occurrence of ruptures at the location of the smallest cross-sectional area, caused by the occurrence of maximum stresses at that location, regardless of the thickness of the components of the structure. The character of the breakage indicates the brittleness of the material at a given location.

## 4. Conclusions

Based on the obtained results, the following conclusions were drawn:Process parameter development for 20MnCr5 with the use of laser power equal to 225 W, an exposure velocity of 600 mm/s, and a hatching distance of 0.10 mm allowed us to obtain samples with a total porosity equal to 0.16%.The total rotation angle for the monolithic sample was 61°, with the loading torque equal to 172 Nm.Almost all tests (instead of PS06) indicated highly consistent results in every tested specimen.To determine the best-obtained properties, in the case of the samples with cellular structures, a torque-to-mass coefficient was introduced. It indicates the best properties of honeycomb structures, which have about 10% smaller torque-to-mass coefficient values than monolithic structures (PM samples).All samples broke in a plane parallel to the powder fusion layers, which is strongly affected by the part’s orientation during the AM process.

## Figures and Tables

**Figure 1 materials-16-01877-f001:**
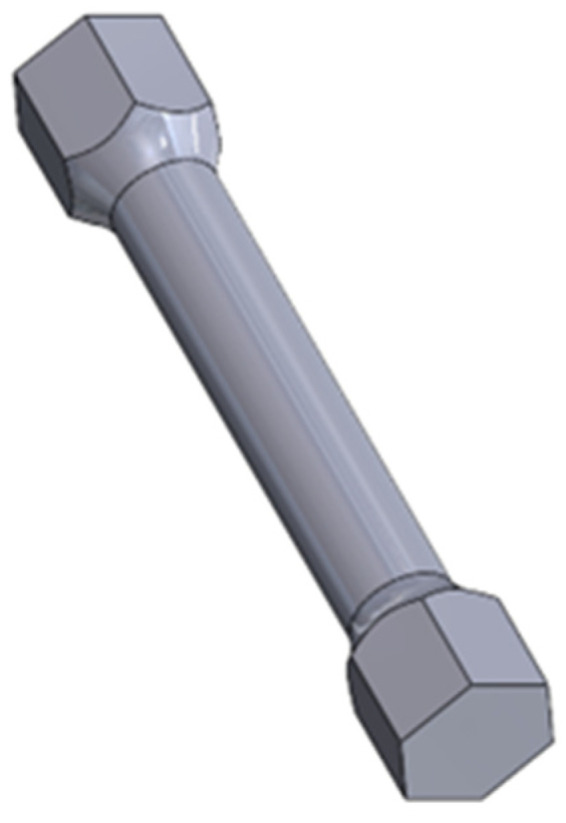
Monolithic specimen (PM_Samples).

**Figure 2 materials-16-01877-f002:**
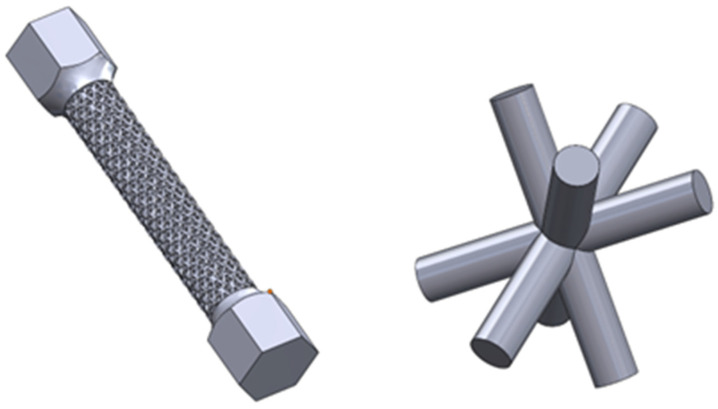
A view of the ‘Kagome-structured’ test sample (**left**) and the single cell (**right**) from which the structure was assembled (PS_Samples).

**Figure 3 materials-16-01877-f003:**
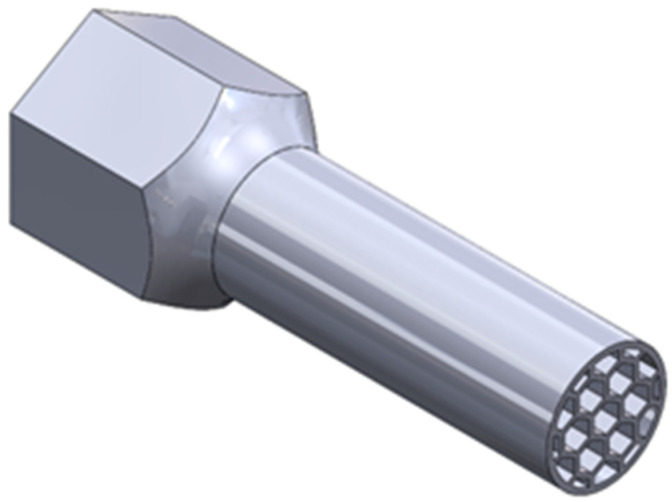
Half on the specimen with a honeycomb cell structure (PH Samples).

**Figure 4 materials-16-01877-f004:**
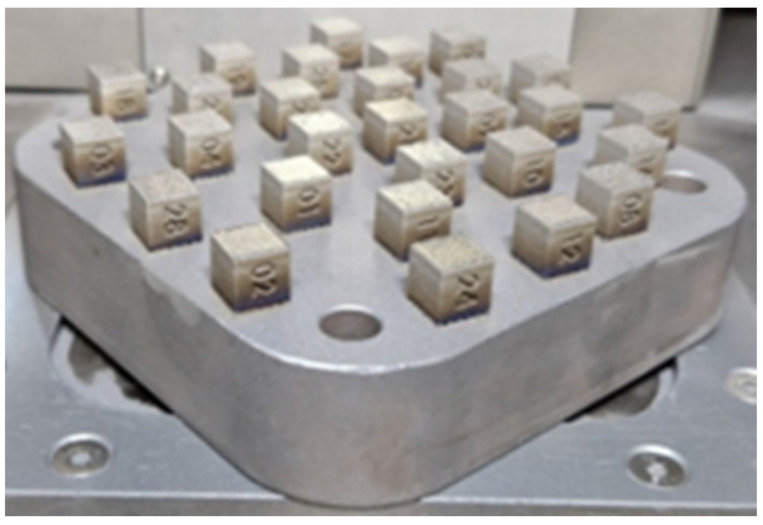
Samples fabricated to determine the most beneficial printing parameters.

**Figure 5 materials-16-01877-f005:**
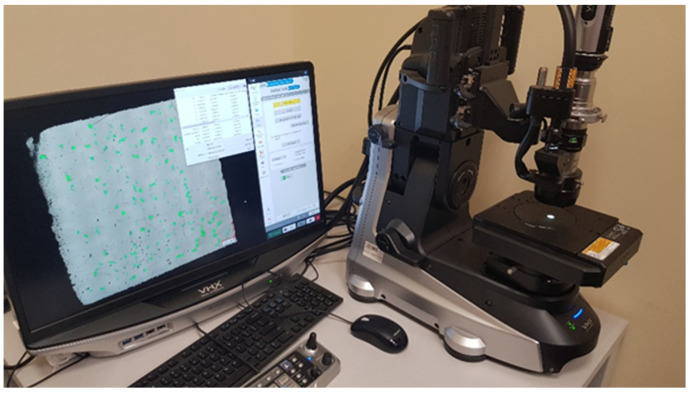
Keyence VHX-7000 digital microscope.

**Figure 6 materials-16-01877-f006:**
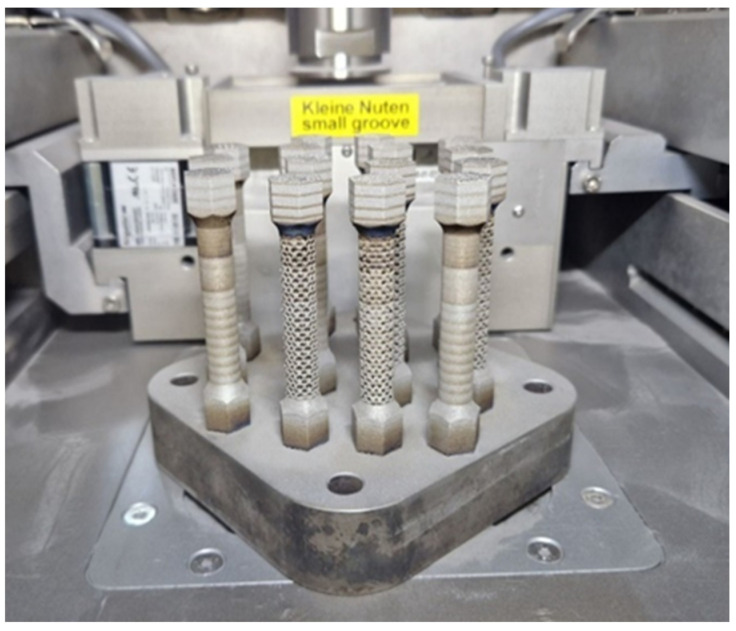
Samples after cleaning from unmelted powder.

**Figure 7 materials-16-01877-f007:**
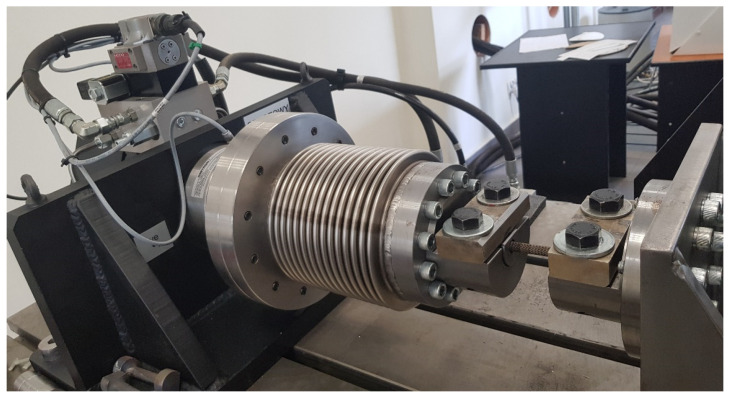
Instron 8802 torsional strength test rig.

**Figure 8 materials-16-01877-f008:**
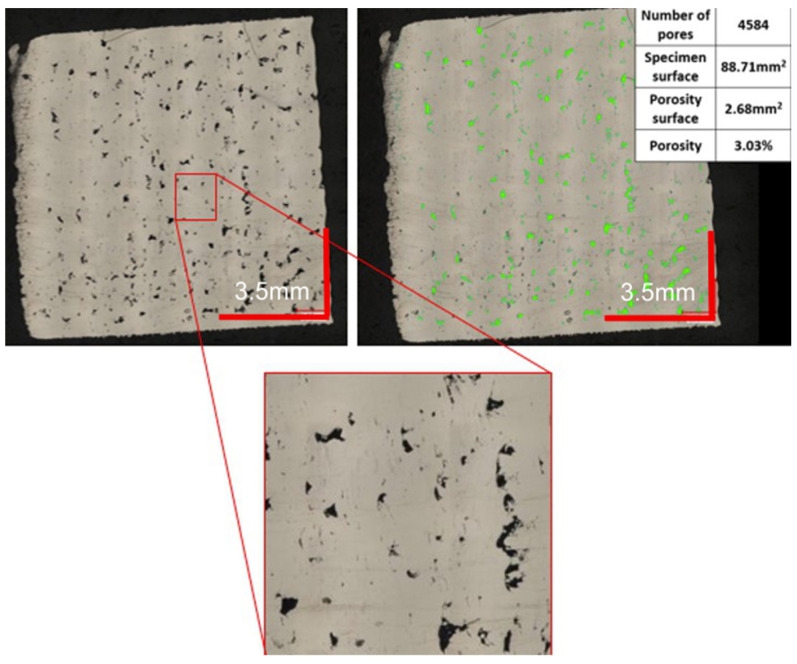
Digital microscope image (**left**) and the same image after porous surface analysis (green areas) on the (**right**).

**Figure 9 materials-16-01877-f009:**
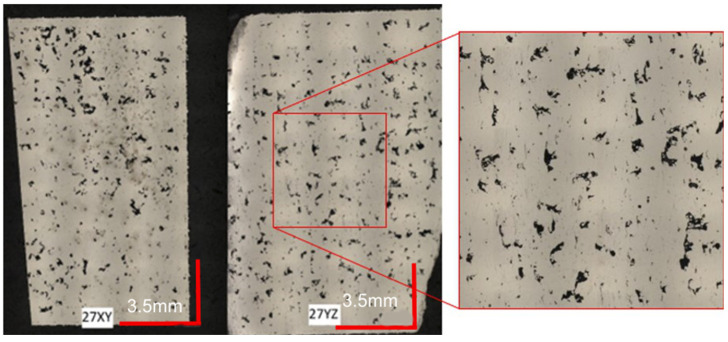
Image of the sample structure produced using 27th parameter group. On the left is the melt plane of a single layer. On the right is a cross-section through all layers.

**Figure 10 materials-16-01877-f010:**
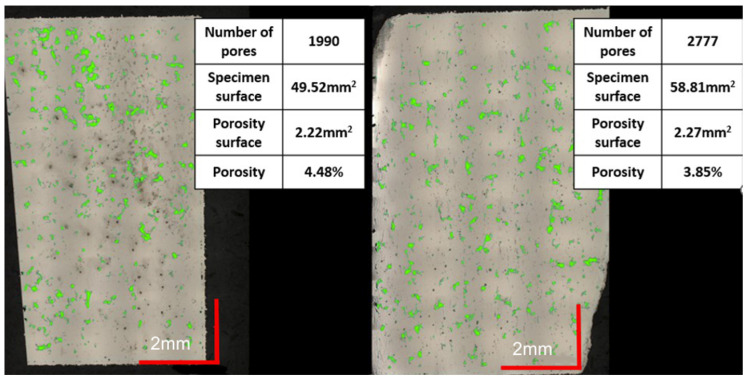
Presentation of porous specimen surface analysis (green color).

**Figure 11 materials-16-01877-f011:**
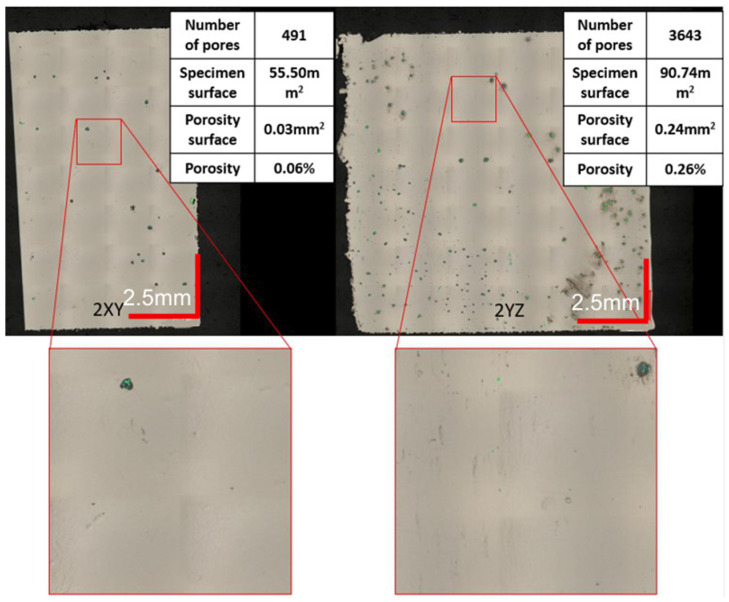
Evaluation of the porosity of a sample produced using 2 groups of parameters.

**Figure 12 materials-16-01877-f012:**
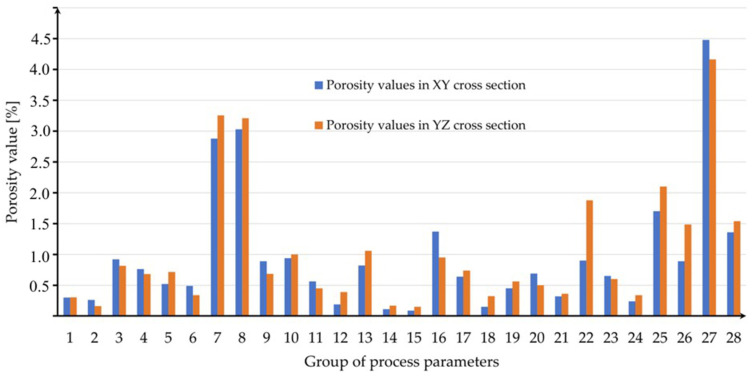
Columnar chart showing porosity in two planes of each tested parameter group.

**Figure 13 materials-16-01877-f013:**
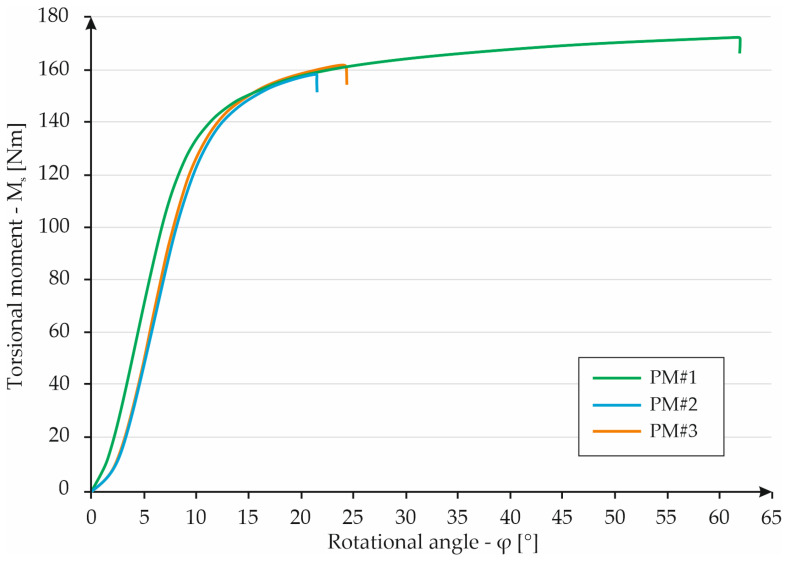
Graph showing the dependence of torque on torsion angle for MS specimens.

**Figure 14 materials-16-01877-f014:**
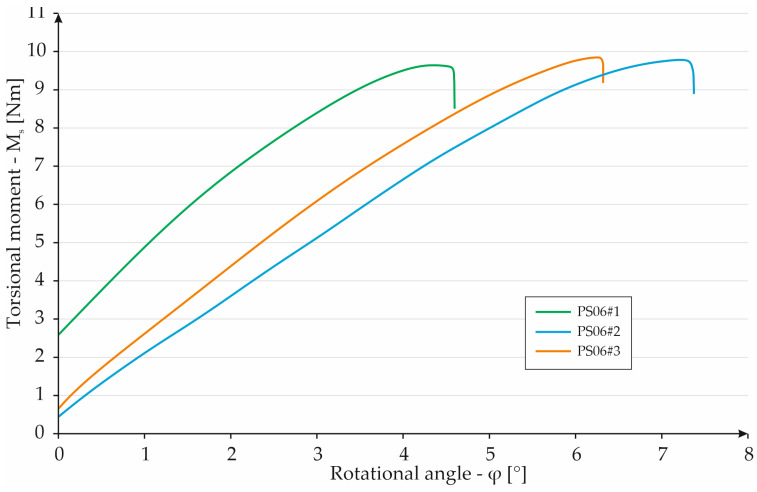
Graph showing the dependence of torque on torsion angle for PS06 specimens.

**Figure 15 materials-16-01877-f015:**
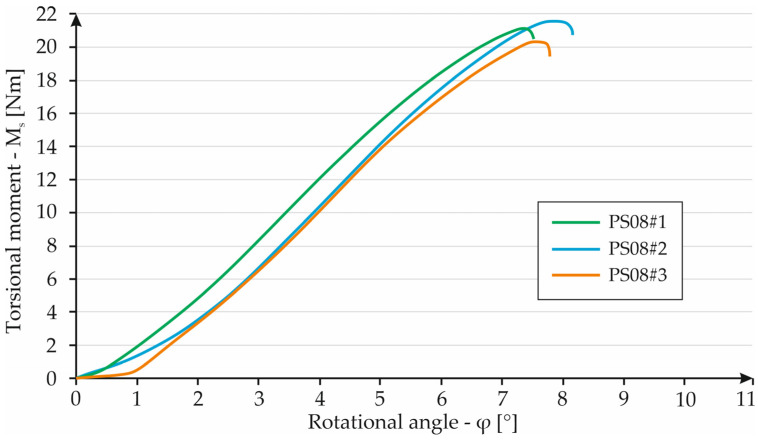
Graph showing the dependence of torque on torsion angle for S08 specimens.

**Figure 16 materials-16-01877-f016:**
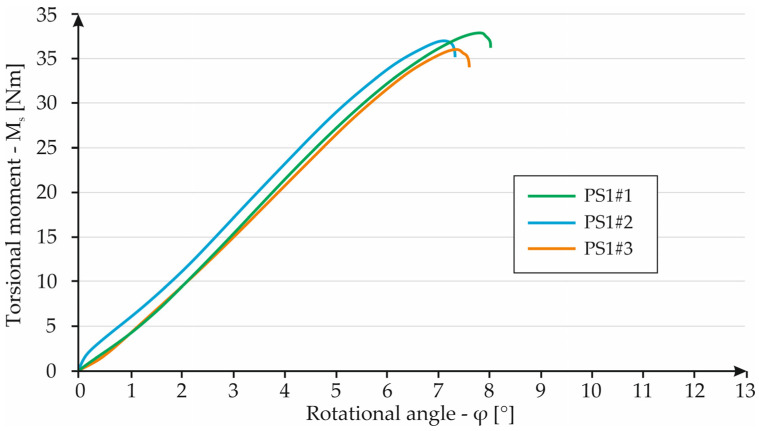
Graph showing the dependence of torque on torsion angle for PS1 specimens.

**Figure 17 materials-16-01877-f017:**
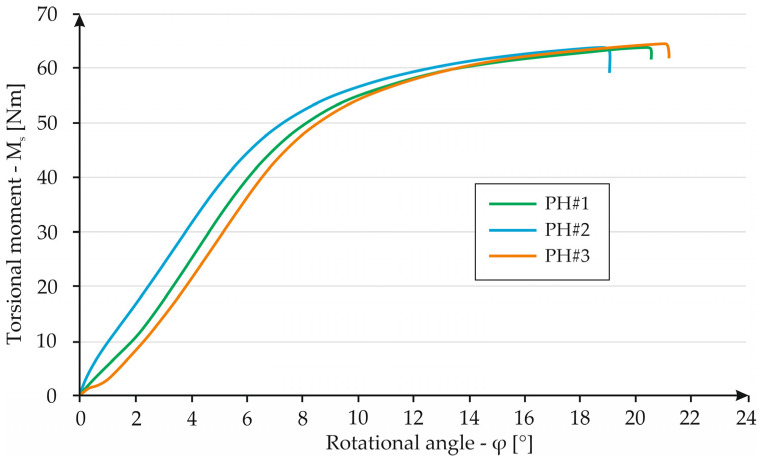
Graph showing the dependence of torque on torsion angle for HS specimens.

**Figure 18 materials-16-01877-f018:**
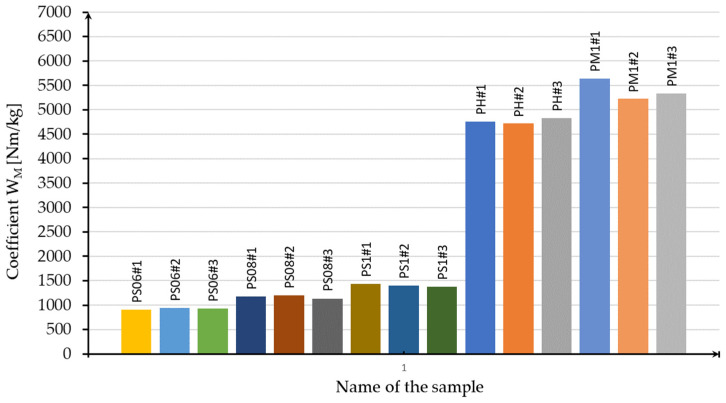
Columnar diagram with the values of the moment-to-torsion angle ratio for all tested samples.

**Figure 19 materials-16-01877-f019:**
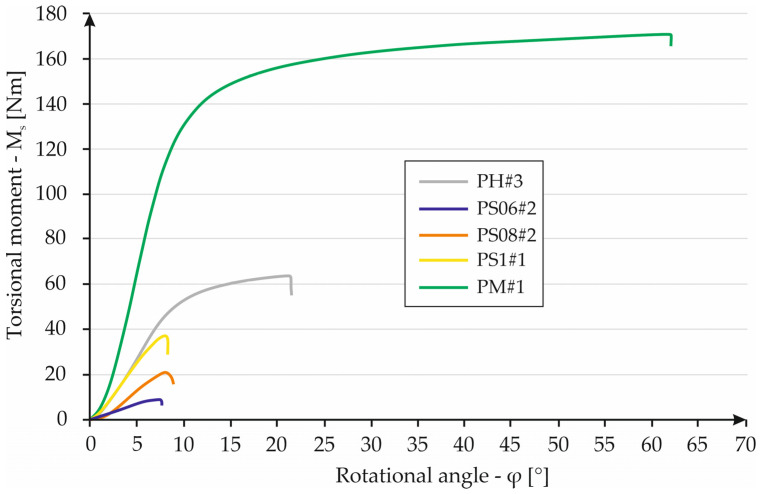
A chart containing the result for each specimen series.

**Figure 20 materials-16-01877-f020:**
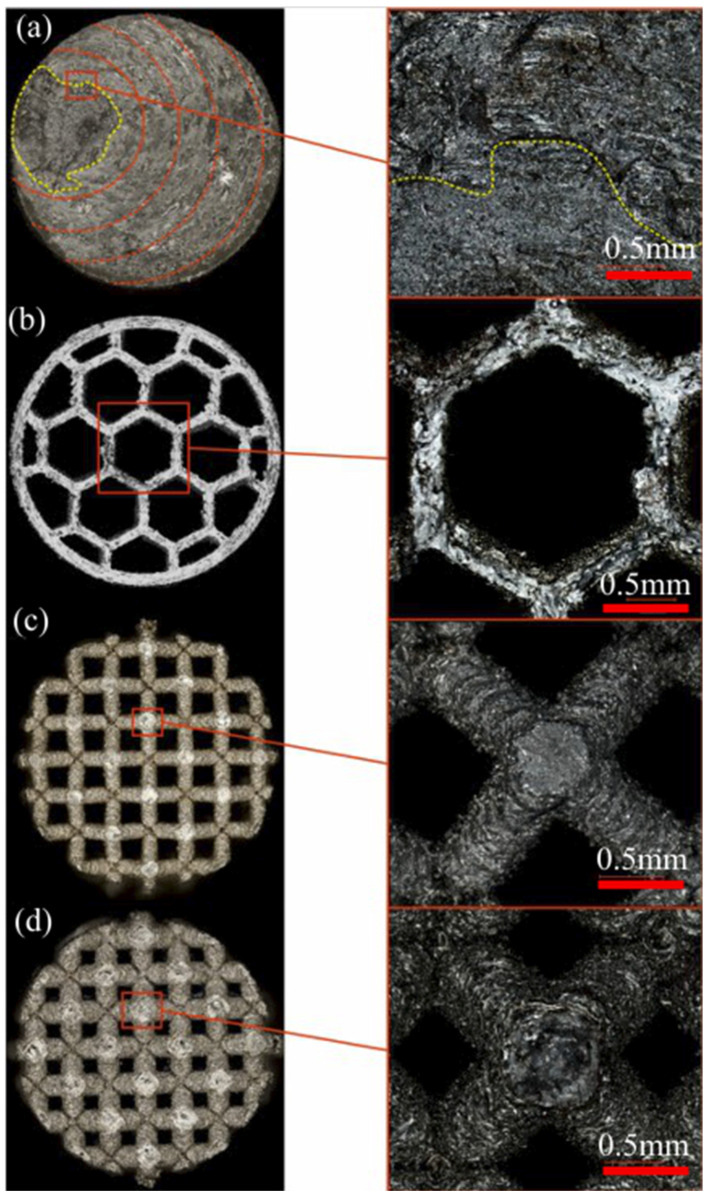
Images of breakthroughs: (**a**) monolithic sample, (**b**) honeycomb structure sample, (**c**) “Kagome” structure sample with 0.6 mm cell diameter and (**d**) “Kagome” structure sample with 0.8 mm cell diameter.

**Table 1 materials-16-01877-t001:** Particle size and chemical composition of 20MnCr5 and 21NiCrMo2 [21] steel powder.

Material	Powder Particles Diameter	C	Si	Cr	Mn	Ni	Mo
20MnCr5	15–45 µm	0.15%	0.19%	0.90%	1.05%	-	-
21NiCrMo2	20–63 µm	0.17–0.23%	<0.40%	0.35–0.65%	0.60–0.95%%	0.40–0.70%	0.15–0.25%

**Table 2 materials-16-01877-t002:** Group of parameters used in the specimen manufacturing process.

Specimen Number	Laser Power (W)	Exposure Velocity (mm/s)	Hatch Spacing (mm)	Energy per Unit Volume (J/mm^3^)	Thickness of Powder Layer [mm]
1	195	600	0.10	108.3	0.03
2	225	600	0.10	125.0	0.03
3	255	600	0.10	141.7	0.03
4	195	700	0.10	92.9	0.03
5	225	700	0.10	107.1	0.03
6	255	700	0.10	121.4	0.03
7	195	800	0.10	81.3	0.03
8	225	800	0.10	93.8	0.03
9	255	800	0.10	106.3	0.03
10	195	600	0.11	98.5	0.03
11	225	600	0.11	113.6	0.03
12	255	600	0.11	128.8	0.03
13	195	700	0.11	84.4	0.03
14	225	700	0.11	97.4	0.03
15	255	700	0.11	110.4	0.03
16	195	800	0.11	73.9	0.03
17	225	800	0.11	85.2	0.03
18	255	800	0.11	96.6	0.03
19	195	600	0.12	90.3	0.03
20	225	600	0.12	104.2	0.03
21	255	600	0.12	118.1	0.03
22	195	700	0.12	77.4	0.03
23	225	700	0.12	89.3	0.03
24	255	700	0.12	101.2	0.03
25	195	800	0.12	67.7	0.03
26	225	800	0.12	78.1	0.03
27	255	800	0.12	88.5	0.03
28	200	1111	0.06	100.0	0.03

**Table 3 materials-16-01877-t003:** Printing parameters used to manufacture cellular specimens.

Parameter and Unit	Value
Power (W)	225
Exposure velocity (mm/s)	600
Hatching distance (mm)	0.1
Layer thickness (µm)	30
Energy per unit volume (J/mm^3^)	125

**Table 4 materials-16-01877-t004:** Registered results for different cell structures.

Specimen	Max. Torque [Nm]	Mass of Structure [g]	W_M_ Coefficient [Nm/kg]
PS06#1	9.65	10.67	904
PS06#2	9.79	10.42	936
PS06#3	9.86	10.64	927
PS08#1	21.09	17.94	1175
PS08#2	21.55	17.93	1202
PS08#3	20.33	17.94	1133
PS1#1	37.96	26.45	1435
PS1#2	37.01	26.48	1398
PS1#3	36.04	26.11	1380
PH#1	63.82	13.40	4764
PH#2	63.76	13.51	4720
PH#3	64.46	13.35	4828
PM#1	172.19	30.52	5641
PM#2	158.38	30.33	5222
PM#3	161.70	30.33	5332

## Data Availability

Not applicable.

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
