# Peer review of "Process Parameter Investigation and Torsional Strength Analysis of the Additively Manufactured 3D Structures Made of 20MnCr5 Steel"

_materials, 2023, doi:10.3390/ma16051877_

Round 1
Reviewer 1 Report
Dear Authors,
I have read the paper with interest, and please find below some comments that I hope can help your acceptance process.
Before listing thecomments, I have a general question: why torsional testing and not standards fatigue or tensile testing? If you have in mind some application, please state, because this can help the better undestanding of the work.
List of Comments:
Line 34 or after line 38: it is suggested to add another general sentence, such us. ’Furthermore, additive technology has recently shown promise for the production of components with alloys that cannot be processed with traditional processes, for example in the case of magnetic FeSi steels with high Si content [ there are further 3 references listed…for references]’’.
I. Tiismus, H.; Kallaste, A.; Vaimann, T.; Rassõlkin, A. State of the art of additively manufactured electromagnetic materials for topology optimized electrical machines. Additive Manufacturing, 2022, 102778.
II. Di Schino, A.; Stornelli, G. Additive Manufacturing: A New Concept For End Users. The Case Of Magnetic Materials. Acta Metallurgica Slovaca. 2022, 28(4), 208 – 211
III. Stornelli, G.; Faba, A.; Di Schino, A.; Folgarait, P.; Ridolfi, M. R.; Cardelli, E.; Montanari, R. Properties of additively manufactured electric steel powder cores with increased Si content. Materials. 2021 14(6), 1489. Doi: 10.3390/ma14061489
Table I: please, change the Table heading: these are not the properties, but the poweder size and nominal chemical composition.
Line 103: please report in table I also the details for 21NiCrMo2 steel. This is for helping the reader to not stop and look for your reference [16]. Moreover, say something more about the reasons for assuming that the same behaviour is expected also for the steel under examination: it is just same equipment? Very similar chemical composition? Same powder sizes? Etc etc add your reasons and more details about the selected methodology.
Table 2: The thickness of the power layer is fixed? Please, assess it if missed.
Line 154: Please, specify what you mean for “typical porosity analysis”. There are a lot of standards and also simpler methods such us Archimedes approaches….please avoid the word “typical”.
Line 154- 156: The image automatic detection of porosities is a “tricky problem”: it depends strongly on calibration and on settings. How many measurements for each area do you have done? Have you considered the mean values? What data dispersion You have found? Please, add these data or give a rationale for convincing that the data are robust and reliable.
Line 167, there is a not necessary dot.
Figg 8, 10 and 11: there are labels that are useless, it Is not possible to read them so it is suggested to add tables with these details, improve the images quality, It is suggested to add an higher magnification detail.
Fig.9: The figure reports the 27th group section and figure 12 its measured porosity. It is only 4,5%? Please, check it.
Table 3: the layer thickness is reported to be 30 microns, but the powder used (from Table 1) is 15-45 microns. How it is possible? If the powder of 45 microns is dragged and removed from the 30 microns layer this fact will result in a heavy porosity because of lack of material not due to the process parameters, I hope you understand and please check and explain better.
Line 305: The authors stated : “This research was funded by the Military University of Technology,” what Military University ? In the world there are more than one….please add the complete references.
Reviewer 2 Report
Comment #1, Page 9 line 215:
From Fig.13, please give some discussion about the torsion angle results of 3 specimens which are quite different between (PM#1) and (PM#2, PM#3).
Comment #2 Page 13
Please provide more theoretical analysis and discussion to explain why the honeycomb structure has better of torque-to-mass coefficient "WM" than Kagome structure.
Comment #3
If possible, could you please provide some pictures and failure analysis results from the broken specimen after torsion test, with some discussion.
Comment #4
In Fig 14., why the curves does not start from Ms = zero like the other figures.
Comment #5, some small points or misspelling are found as follows:
Page 8 line 185
Misspelling is found: 'The are significant differ-"
(Did you mean "They are" or "There are"?)
Page 5 line 134-136
Is there any different of shape of single cell beside the diameter in the samples S06, S08, S1? Why do you use the word 'bar' in S06 and S08 and use the word 'rod' in S1?
In Fig 18, should the column of PH be in position after PS like in the table4 so that it will be the same in order.
Round 2
Reviewer 1 Report
Good Work